



Atmospheric
Measurement
Techniques

# Support vector machine tropical wind speed retrieval in the presence of rain for Ku-band wind scatterometry

**Xingou Xu[1] and Ad Stoffelen[2]**

[1]The CAS Key Laboratory of Microwave Remote Sensing, National Space Science Center,
Chinese Academy of Sciences, Beijing, 100190, China
[2]Satellite Observations, Royal Netherlands Meteorology Institute, De Bilt, 3730 AE, the Netherlands

**Correspondence:** Ad Stoffelen (ad.stoffelen@knmi.nl)

**Abstract.** Wind retrieval parameters, i.e. quality indicators and the two-dimensional variational ambiguity removal (2DVAR) analysis speeds, are explored with the aim to improve wind speed retrieval during rain for tropical regions. We apply the well-researched support vector machine (SVM) method in machine learning (ML) to solve this complex problem in a data-oriented regression. To guarantee the effectiveness of SVM, the inputs are extensively analysed to evaluate their appropriateness for this problem, before the results are produced. The comparisons between distributions and differences between data of rain-contaminated winds, corrected winds and good quality C-band winds illustrate that the rain-distorted wind distributions become more nominal with SVM, hence much reducing the rain-induced biases and error variance. Further confirmation is obtained from a case with synchronous Himawari-8 observation indicating rain (clouds) in the scene. Furthermore, the estimation of simultaneous rain rate is attempted with some success to retrieve both wind and rain. Although additional observations or higher resolution may be required to better assess the accuracy of the wind and rain retrievals, the ML results demonstrate benefits of such methodology in geophysical retrieval and nowcasting applications.

## 1 Introduction

It is well known that the structure of the atmosphere and ocean depends on the motions driven by radiation affecting the redistribution of heat. The circulations imply wind convergence to elevate water vapour from the ocean surface that then forms clouds and rain, while rain, in turn, causes downdrafts. These interactions of the air and the ocean underneath connected by the basic mass, momentum and energy equations involving winds, heat and moisture are vital for understanding the Earth system (Gill, 1982). In the tropics, the resolution of moist convection is key for improving earth system simulations (Bony et al., 2015).

Observed ocean surface wind fields (OSWs) CE1 are essential to investigate such processes and related applications. An efficient method of acquiring large coverage and good quality OSW is by using the retrievals from scatterometers with an application history of up to 40 years (Lipes et al., 1979 TS1; Stoffelen et al., 2019). Scatterometers are real aperture radars providing stable and accurate normalized radar cross sections (NRCSs) of the wind-roughened ocean surface in different azimuthal directions from oblique incidence angles. The winds are then obtained in a maximum likelihood estimation method (MLE) from the measured NRCSs within a wind vector cell (WVC) with reference to a geophysical model function (GMF). Generally, a WVC is a square of the size $25\,\text{km} \times 25\,\text{km}$, and GMFs are empirical models mapping NRCSs from scatterometers in different frequencies, polarizations and observing geometries to winds.

Rain products provide another important information for air–sea interaction. In the Global Precipitation Mission (GPM), one of the core instruments is the dual-frequency precipitation radar (DPR) working at Ku and Ka bands in nadir-looking mode. Rain is then obtained by relating the radar cross sections to a chosen distribution of precipitation particles. Meanwhile, rain products from infrared observations are also widely used, for example, rain rates from the

Spinning Enhanced Visible and Infrared Imager (SEVIRI) aboard the Meteosat Second Generation (MSG) satellite, which is derived by considering retrieved cloud-condensed water path (CWP), particle distribution and cloud thermodynamic phase (Wolters et al., 2011). Both rain products are good references for rain in Ku-band wind scatterometry (Xu et al., 2020a), though the high spatial and temporal variability of rain generally challenges small collocation errors and high correlation between instantaneous rain data sets (e.g. Liu et al., 2020).

Combined retrievals of wind and rain are generally applying synchronous passive measurements from radiometers for rain in the scatterometer case (Stiles and Dunbar, 2010), while in addition to rain, winds are retrieved in GPM researches (Li et al., 2004). Radiometer winds are of coarser spatial resolution and are not adept for wind direction retrieval, which would require the third and fourth Stokes parameters that are now generally obtained in a low signal-to-noise ratio (SNR). Scatterometers are not specifically designed for acquiring precipitation profiles. When rain clouds affect the observations, the winds obtained from a wind GMF will deviate from the truth, resulting in biases in the retrieved wind and an increased retrieval residual, called MLE. Since rain is spatially more heterogeneous than winds are, rain can be captured and estimated in the NRCS set within a WVC. Considering the distances of NRCS observations to the wind GMF, the retrieved wind and with the reference to rain observations from the Tropical Rainfall Measuring Mission (TRMM) Precipitation Radar (PR), wind and rain may be segregated (Owen and Long, 2011; Draper and Long, 2004). Furthermore, the heterogeneous rain within a WVC can be depicted from indicators applied in scatterometer quality control (QC) (Portabella and Stoffelen, 2002; Lin and Portabella, 2017). Joss is a recent indicator developed for tropical regions for rain screening, which has been verified to correlate well with rain for Ku-band scatterometers (Xu and Stoffelen, 2020; Xu et al., 2020a). From a conceptual point of view, the MLE identifies the WVC NRCS sets that do not follow the wind GMF. Two main reasons have been identified for such discrepancy, which are (1) enhanced wind variability and (2) rain. Fortunately, collocated operational C- and Ku-band observations are available when, due to the longer wavelength at the C band (about 5 cm) than Ku band (about 3 cm), standard QC, based on MLE, rejects 10 times more Ku-band than C-band winds, i.e. about 5 % of its observations. Hence, specifically in tropical regions, the accepted C-band winds can be used to verify their Ku-band collocations, which helped to develop the Joss TS2 indicator and verify the performance of the other Ku-band QC indicators. In addition, extreme convergence and divergence in C-band winds have been related to tropical moist convection and rain, where convergence proceeds rain by about 30 min, while extreme divergence occurs simultaneously with rain in convective downdrafts for C-band winds, hence illustrating the physical integrity of C-band winds in the presence of rain.

C-band rejections correspond to the most extreme variability in WVCs, including wind gradients induced by heavy precipitation downdrafts (King et al., 2007). The different rain signatures in C- and Ku-band scatterometers can cast a light on developing methods for correction of the rain-affected winds in Ku-band scatterometer retrievals by referring to their C-band collocations. Particularly, the combination of MLE and Joss appears promising to segregate wind variability and rain effects in Ku-band retrievals.

To derive the complexly associated wind and rain information referred to above, machine learning (ML) may prove to be a powerful tool, which can be applied with knowledge of the validity of the underlying principles (Reichstein et al., 2019). In fact, ML methods have long been well researched in wind scatterometry (Thiria et al., 1993; Stiles and Dunbar, 2010). For common roughness conditions, it cannot exceed the performance of GMF-based methods (Cornford et al., 1999), but ML may be effective in rainy conditions. Among the ML methods, support vector machine (SVM) is one based on the Mercer theorem, complements the empirical risk minimization with Vapnik–Chervonenkis (VC) confidence, infers statistical relations without a priori distributions and gives no regional minimum (Vapnik, 1998). It can establish an information space based on the training set and if the data applied in training are well representative of the problem; it also requires fewer samples than other ML methods. Aside from that, SVM already provides good results in rain rate estimates (Kumar et al., 2021).

In this research, SVM is applied for wind correction of rain-affected winds of Ku-band scatterometers, considering quantified rain and rain effect information captured in the QC indicators of Ku-band observations. The GPM rain products and collocated accepted winds from C-band products are used as references. When this SVM model has been established, without C-band collocations, the rain-contaminated winds can be corrected with Ku-band winds and their QC indicators alone. First, in the Method section, the underlying principles of the problem of rain signatures in scatterometry are addressed in detail with a brief on error requirement for assimilation application before data description. Then, in the experimental part, results for the testing set, not applied in the training procedure, demonstrate a minimum mean difference of $-0.12$ m/s at about 8 m/s and a largest difference of $-3.25$ m/s at about 14 m/s Advanced Scatterometer (ASCAT) speed. The distribution of the corrected winds and the scatter plots against C-band winds are inspected, with a check on wind differences in each wind speed bin of the original and corrected winds against ASCAT winds, proving the more unbiased and symmetric error of the corrected set, illustrating the advantage of applying SVM. The similarity of the corrected distribution with the references provided from collocated ASCAT winds and the reduced mutual differences indicates that to a certain extent the local (WVC) wind scales are recovered by the SVM corrections. Results suggest that the method resolves the heterogeneity induced by rain clouds

in MLE and Joss with the settings of the proposed SVM. Furthermore, a case without rain collocations, and thus not involved in deriving the corrections, is provided as a case study for verification, where simultaneous images from the Himawari-8 provide a concrete view of the rain clouds in the scene.

In the discussion part, rain labelling and regression SVMs are established with the same inputs, attempting rain estimation from scatterometer winds by employing SVM. The rain identification accuracy is 72 % for the independent test set not applied in the training procedures. While for rain rate estimation, the correlation coefficient of SVM rain with GPM products achieves 0.47 for the independent testing set. An analysis of the uncertainties in the SVM model and possible improvements in the rain estimation procedure are also discussed. The corrected winds increase the global wind coverage and, in synergy with the rain information provided, benefit nowcasting applications (Majumdar et al., 2021). This research illustrates an example of complex data-driven ML methods, complementary to traditional methods in complex problems, which motivates and demonstrates the adhibition of the ML method in meteorological applications.

## 2  Method

Research on observation errors, i.e. the deviations from the truth, together with the monitoring information obtained from differences between scatterometer winds and models, support numerical weather prediction (NWP). Among the errors, undetermined geophysical dependencies including rain effects are to be corrected to better understand model biases (Stoffelen et al., 2021), while it cannot be achieved by a first-order correction. Apart from this, the control variables, defining multivariate background errors and correlated errors between variables are modelled by linear regression (Descombes et al., 2015). Also, the 3D-Var and Kalman filter assumes linear or quasi-linear and Gaussian features in observation operator and error distributions, respectively, when 4D-Var considers additional dynamical constraints in the time dimension (Parrish and Derber, 1992; Courtier et al., 1994). Hence, linearized Gaussian or quasi-Gaussian errors are vital for the assimilation of observations. We seek to address and correct biases in Ku-band scatterometer wind retrievals due to rain. In the following part, first, the complex rain signatures in wind scatterometer observations are analysed, demonstrating non-Gaussian error features before the principles of SVM are introduced.

### 2.1  Rain characteristics in MLE, Joss and the fractal parameter $\alpha$

When compared to the C-band winds that are of good quality (accepted), collocated Ku-band QC-rejected WVCs in tropical regions are affected by rain due to the shorter observing

wavelength (Xu and Stoffelen, 2020). The wind QC is determined by QC indicators, and the indicator widely applied in operational wind products is the MLE residual obtained through wind inversion. Using all $N$ (number of) NRCS measurements obtained within a WVC, the maximum livelihood estimation procedures are applied for wind retrieval. The MLE residual is a normalized Euclidian distance to the cone determined by GMFs (Stoffelen and Anderson, 1997):

$$\text{MLE} = \frac{1}{N} \sum_{i}^{N} \frac{\left(\sigma_i^o - \sigma_{\text{sim}_i}\right)^2}{\left(K_{\text{pi}} \cdot \sigma\right)^2}, \tag{1}$$

where $\sigma_i^o$ is the $i$th NRCS of the $N$ NRCSs within a WVC, $K_{\text{pi}}$ is a dimensionless constant determined by instrument noise, and $\sigma_{\text{sim}_i}$ is from a wind GMF indexed by observing geometry and the local wind vector TS3. Before wind inversion, NRCS are well calibrated for instrumental as well as GMF uncertainties that are generally small ($\sim 2\%$) and are reproducible or systematic. NRCS calibration and GMF bias term uncertainties lead to wind speed probability density function variations. Errors in the harmonic terms of the GMF may lead to wind direction errors, and in systematic wind speed errors that have associated wind direction errors, and vice versa (Portabella and Stoffelen, 2002). During the two-dimensional variational ambiguity removal (2DVAR) procedure that optimizes wind vector selection (Vogelzang and Stoffelen, 2011 TS4), essentially the WVC MLE associated with the selected direction is determined. At the same time, the 2DVAR low-pass-filtered analysis winds, which are here referred to as 2DVAR winds, are calculated. When rain affects the NRCS, the GMF does not represent the NRCS measurements well, as rain effects are not considered in the wind GMFs (Stoffelen, 1998). Therefore, this part of the GMF error due to missed or incompletely modelled rain processes generates errors of a class that cannot be eliminated by calibration and induces deviation of error distributions from the well-calibrated random Gaussian shape. Note that the Royal Netherlands Meteorological Institute (KNMI) QC flag is based on MLE values, and in the Ku-band rejections and C-band acceptances in tropical regions, the rejections are mainly caused by rain. Hence, MLE values of the 2DVAR selected Ku-band wind can be related to rain effects that alter the amplitudes of NRCSs.

However, at the same time, the 2DVAR winds do not use QC-flagged WVCs and are hence not affected by local disturbances introduced by rain. The wind speed correction procedure employed here hence does not change the 2DVAR analysis field, nor the selected wind direction at the rain-affected WVCs obtained during the elaborate 2DVAR multiple solution scheme (MSS) (Vogelzang and Stoffelen, 2018). The rain effect is estimated by the wind speed difference of the 2DVAR analysis wind speed $f$ and the selected observational wind speed $f_s$, corresponding to the wind direction obtained by 2DVAR (Xu and Stoffelen, 2020):

$$\text{Joss} = f - f_s. \tag{2}$$

Note TS5 that the 2DVAR winds are low-pass filtered and of relatively coarse resolution, ignoring rain-affected WVCs through MLE-based QC (Vogelzang, 2007). Since the spatially heterogeneous tropical rain clouds are generally of smaller spatial scale than a WVC, rain effects in the 2DVAR analysis winds can be ignored and taken as the *true* winds (Stoffelen and Vogelzang, 2018 TS6). Hence, Joss values can screen and eliminate false alarm rate (FAR) for MLE-based QC results for Ku-band wind products after 2DVAR processing, indicating rain information (Xu et al., 2020b TS7).

Usually rain clouds will cause negative Joss for wind speeds below 15 m/s. A WVC is usually partially heavy rain, and since Ku-band rain saturates around 18 m/s, hereafter the parameter for area fraction $\alpha$ for Ku-band winds can be expressed as

$$\alpha = \frac{\text{Joss}}{f - 18}. \tag{3}$$

As 18 m/s winds cannot be distinguished from rain and to allow rain sensitivity, the rain effect correction set is limited to

$$\text{Joss} > 0.33 f - 5 \tag{4}$$

for TS8 retrieved 2DVAR speed smaller than or equal to 11 m/s. For 2DVAR wind speed larger than 11 m/s, the set is limited to $\text{Joss} < -1.33$ TS9 (Xu and Stoffelen, 2021). Then the negative values of $\alpha$ corresponding to positive Joss when wind speeds are smaller than 18 m/s can be due to effects of local variance of the ocean surface. Larger wind speed than 18 m/s and positive Joss may happen when both rain and winds are large in the scene. For tropical rain, this practically only occurs in hurricanes but has not yet been investigated with respect to Joss in the criterion above. Thus, this parameter can provide relative information of rain within the WVC from 2DVAR residuals.

Enhanced wind variability enhances MLE due to beam collocation errors. In particular, extreme wind convergence and divergence are associated with heavy rain (King et al., 2007). The wind variability associated with heavy precipitation may enhance the wind speed, just like rain does at the Ku band, but which has been investigated by comparing the 2DVAR winds with ASCAT winds. ASCAT winds are equally sensitive to wind speed variations at the surface but much less sensitive to rain cloud scattering effects. Hence, the effect due to amplitude alternations for a single NRCS in a tropical scene with rain clouds can be obtained by the rain screening ability of Joss.

From the above contents and equations, rain effects can be represented by MLE, Joss, and the observational wind in the Ku-band retrieval, while the 2DVAR analysis wind provides information on rain sensitivity. In this research, for the C-band QC-accepted and Ku-band-rejected WVCs, after the FAR set is eliminated, the Ku-band WVCs are collocated with rain rates from GPM products. Then MLE, Joss values

and the 2DVAR winds and observational winds are applied as inputs to the SVM model, with the training destination set as the collocated C-band winds. In the established model, corrected winds closer to the observed C-band winds may be obtained for rain-affected Ku-band WVCs, by eliminating non-Gaussian errors within a WVC caused by rain. Moreover, the SVM model, when established, could be applied for Ku-band rejections.

## 2.2 The principle of SVM regression

The SVM regression procedures map input vectors to a space of higher dimension before the regression is conducted. When the mapping is obtained and thus described by kernel functions determined from the training sets, non-linear features are linearized. This provides a possibility for solving problems that are non-convex and difficult to solve in the original input space, as well as linearizing intricate relations. Specifically, during the training procedure, weights for the input vectors in the training set in the mapped space is determined, and the corresponding support vectors (SVs) can be identified by the values of corresponding weights, while the weights are applied to scale similarities with other vectors in the training set. On the other hand, they are obtained by minimizing distances with the targets of the training vectors. Moreover, the similarity is measured between the kernel function mapped inputs. In this way, it allows the data involved in training to embody the underlying model in a space that facilitates information extraction. Furthermore, L2-normalized distance minimization is achieved by an objective function expressed as the distances between the vectors in the training sets to the plane fixed by the weighted support vectors in the mapped space (Vapnik, 1998).

The employed kernel functions are linear, generally polynomial or Gaussian radial basis functions (RBFs). Among them, the RBF, or the Gaussian kernel, is superior in unlimited dimension mapping and easier in hidden parameter setting. For RBF, the similarity between a vector $x$ and the selected support vector $l^{(1)}$ is expressed as (Vapnik, 1998; Smola and Schölkopf, 2004)

$$f_1 = \exp(-\frac{||\left|x - l^{(1)}\right||^2}{2\sigma^2}), \tag{5}$$

where $\sigma$ is the scale parameter weighting the similarity of $x$ and $l^{(1)}$. And the larger the value of $\sigma$ is, the more $x$ and $l^{(1)}$ can be taken as similar. If the L2 distance (Euclidean distance) is applied,

$$f^1 = \exp(-\frac{\sum_{j=1}^{n}(x_j - l_j^{(1)})^2}{2\sigma^2}). \tag{6}$$

When $\boldsymbol{\theta_i}$ are weights, $\boldsymbol{y^{(i)}}$ is the target value corresponding to $x_i$, the objective function can be expressed as TS10

$$\min_\theta (C \sum_{i=1}^{m} \boldsymbol{y}^{(i)} \mathrm{cost}_1 (\theta^T \boldsymbol{f}^{(i)}) + \left(1 - \boldsymbol{y}^{(i)}\right) \mathrm{cost}_0$$

$$\times \left(\theta^T \boldsymbol{f}^{(i)}\right) + \frac{1}{2} \sum_{j=1}^{m} \theta_j^2), \tag{7}$$

where TS11 $C$ is the relaxation coefficient and the L2 distance (Euclidean distance) is applied as the cost functions $\mathrm{cost}_1$ and $\mathrm{cost}_0$ (Smola and Schölkopf, 2004; Chang and Lin, 2011 TS12).

## 3 Data and experiments

### 3.1 The expression of rain in wind retrieval parameters

The representativeness of the data sets from which the featured SVs are obtained is vital in the SVM procedure. In this research, the C- and Ku-band collocations of scatterometer winds are from the Advanced Scatterometer-A (ASCAT-A) and ASCAT-B aboard the Meteorological Operational Satellite Program of Europe (MetOp) series and the scatterometer aboard the Scatsat-1 satellite (OSCAT-2) respectively. TS13 Then the ASCAT-A, ASCAT-B and OSCAT-2 L2 wind products are from the Ocean and Sea Ice Satellite Application Facility (OSI SAF) of the European Organization for the Exploitation of Metrological Satellites (EUMETSAT), over a period from October 2016 to January 2019. The WVC sizes are $25\,\mathrm{km} \times 25\,\mathrm{km}$ on the Earth's surface. Where the OSCAT-2 Ku-band winds are sea surface temperature (SST)-corrected sweet swath WVCs with better NRCS azimuth diversity than the nadir and edge swath (Portabella, 2002). The collocation time lag is within 30 min (min) with the spatial distances between ASCAT and OSCAT-2 WVC centres less than 12.5 km. While the background winds are from the European Center for Medium-range Weather Forecasts (ECMWF), the 10 m 3-hourly forecast 0.125° winds are used. GPM rain products used here are the version 5 0.1°-gridded Integrated Multi-satellitE Retrievals for GPM-F (IMERG-F) (Huffman et al., 2018) within a time difference to OSCAT-2 WVCs of 4.8 min. Furthermore, rain products are area weighted over the OSCAT-2 WVCs to obtain WVC-representative rain rates (Xu et al., 2020b). Finally, for validation, the images of the 11th band (medium infrared, MI, 8.6 µm) with 2 km resolution in the tropics are also used for reference (Japan Meteorological Agency, 2015).

Figure 1 plots the 732 614 collocated wind speeds in the ASCAT-A-accepted and OSCAT-2-accepted set (QC-I set) in (a), corresponding MLE values of OSCAT-2 in (b), Joss in (c) and collocated rain rates in (d) and (e). Figure 2 shows the same plots for the ASCSAT-A accepted winds but now for rejected OSCAT-2 collocations (QC-II), after that the false

alarms in the KNMI OSCAT flags were eliminated by Joss (FAE), with 9339 WVCs (Xu et al., 2020b).

In Fig. 1a, we note that observed wind distributions from ASCAT and OSCAT-2 are similar, while in Fig. 2a, the Ku-band winds are much elevated with respect to ASCAT and clearly suspect, as the ASCAT wind distribution appears nominal and similar to that in Fig. 1a. The MLE values are mostly nominal and distributed over the bins under 10 in Fig. 1b, while typical values are very large and around 50 in Fig. 2b. For comparison, in panel (c) of Fig. 1, Joss values are small with values close to 0, wherein Fig. 2 values are typically 4 m/s. Comparing panels (d) and (e) in both figures, there is little rain in QC-I, while rain is dominant in Fig. 2, consistent with both the elevated MLE and Joss values. Also, in Fig. 2e, the criterion of Joss in the FAE set can be observed from its upper limit.

We note from Figs. 1 and 2 that rain casts effects on OSCAT-2 data, while collocated ASCAT winds remain of acceptable quality. The winds distorted by rain (clouds) are clearly segregated by the FAE, resulting in a deformed speed distribution, as well as much elevated MLE and Joss, that all can be potentially related to WVC rain rate.

### 3.2 SVM for Ku-band wind correction in rain

For the correction of rain effects a SVM model is established, where the inputs are determined by the wind–rain-related parameters, as described in the previous sections. Specifically, the inputs and outputs are in Table 1.

The SVM tool from *sklearn* is applied, which is based on the *libsvm* to realize the procedure described in Sect. 2 for SVM (Chang and Lin, 2011). In total, there are 18 528 WVCs obtained from FAE in OSCAT-2 collocations for ASCAT-A and ASCAT-B together. Among them, 70 % (12 969 WVCs) are used in training and 30 % (5559 WVCs TS14) for testing or validation. Note that the testing set is not applied in the training procedure.

## 4 Results and validation

### 4.1 Results

Starting from the large input biases illustrated in Fig. 3a, typically 5 m/s, Fig. 3 shows the corrected winds against the accepted winds from ASCAT-A and ASCAT-B for the training set in (a) and the validation set in (b), while in (c) and (d), the observational winds and 2DVAR winds of OSCAT-2 are also plotted against ASCAT winds. Some of the corresponding statistics are listed from (a) to (d) in Table 2.

As can be seen from Fig. 3a and b, and from the corresponding values in Table 2a and b, the testing set exhibits similar statistics to the training set for wind speed correction established by SVM. Note that most of the QC-II FAE wind speeds are distributed from about 4 to 14 m/s, which is typical for rain clouds in moist convection (Xu and Stoffe-

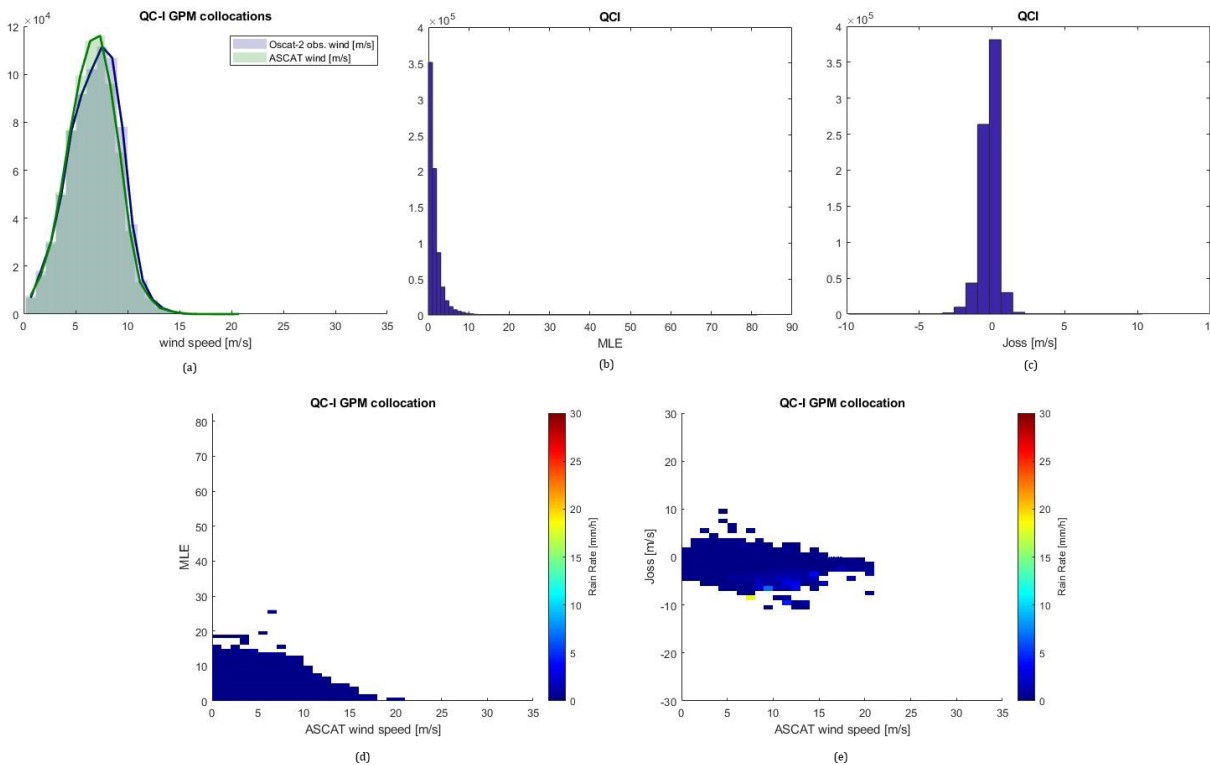

**Figure 1.** Collocated wind speed distributions in the QC-I set (**a**), corresponding MLE distribution of OSCAT-2 (**b**), Joss (**c**), collocated rain rates with reference to MLE (**d**) and Joss (**e**).

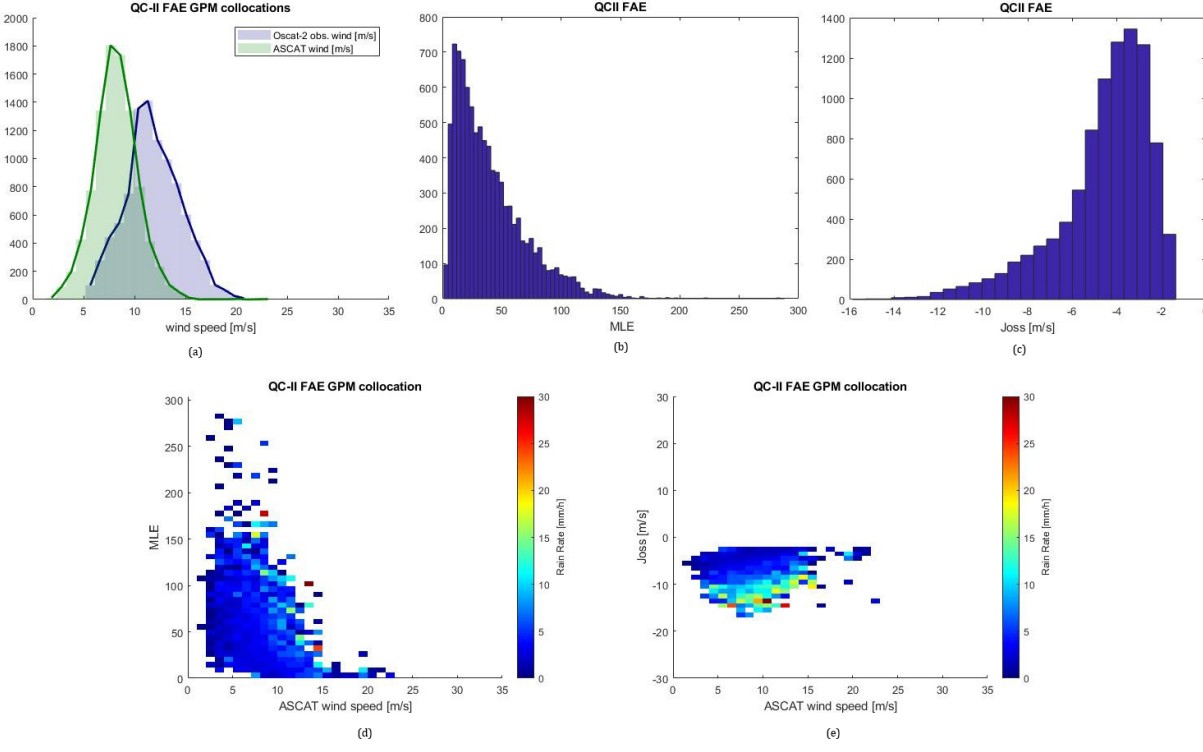

**Figure 2.** Collocated wind speed distributions in the QC-II FAE set (**a**), corresponding MLE distribution of OSCAT-2 (**b**), Joss (**c**), collocated rain rates with reference to MLE (**d**) and Joss (**e**).

**Table 1.** Inputs and outputs for the SVM of wind speed correction.

| Inputs | Output | Values of output for training |
|---|---|---|
| OSCAT-2 MLE in dB<br>$\alpha$ from OSCAT-2 WVC<br>OSCAT-2 2DVAR speed<br>OSCAT-2 observational speed | Corrected OSCAT-2 wind speed [m/s] | ASCAT wind speed [m/s] |

**Figure 3.** The corrected winds against accepted winds from ASCAT-A and ASCAT-B for the training set **(a)** and validation set **(b–d)**, where **(b)** the corrected, **(c)** the 2DVAR and **(d)** observational OSCAT-2 wind speed against ASCAT wind speed are depicted.

len, 2020). For speeds in this range, the largest differences of mean values with the bin centre values are −3.69 and −3.25 m/s at about 14 m/s ASCAT speed for the training and testing set, respectively. Then the bias value decreases as wind speed decreases and, for both sets, reaches a minimum at about 8 m/s of −0.15 and −0.12 m/s. Then the bias increases with decreasing wind speeds to 1.72 and 1.79 m/s at about 4 m/s. This trend is consistent with the SDD, with the smallest SDD of 0.87 m/s for both sets at about 7 m/s. The consistency of the training set and testing set indicates the stability of the SVM model established. Besides, it is noteworthy that there is a sign change for these speed differences, suggesting an excessive speed range suppression

for wind speeds both lower and higher than around 8 m/s, respectively. This trend also exists in Fig. 3c and d of the observational and 2DVAR wind against ASCAT winds, as seen from the curvature of the red lines representing mean bin values, though they are generally smaller and larger than the ASCAT wind speed for the 2DVAR and observational speed, respectively, while the distances are larger in absolute values for the observational winds. This is consistent with the fact that the OSCAT 2DVAR wind filters the details of the local wind changes, ignoring wind variability due to rain that is captured by the C-band observations of good quality at finer resolutions. We further note that Fig. 3 and Table 2 are based on a conditional binning of ASCAT winds, while ASCAT

**Table 2.** Corresponding mean and standard deviation of difference (SDD) statistics to Fig. 3a–d. TS15

| **(a)** Corrected winds in the training set | | |
|---|---|---|
| ASCAT-A and ASCAT-B wind speeds [m/s] | Mean values of the corrected winds [m/s] | SDD between the corrected and ASCAT winds [m/s] |
| 4.14 | 5.86 | 1.11 |
| 6.21 | 6.95 | 0.91 |
| 8.28 | 8.13 | 0.93 |
| 10.34 | 9.54 | 1.21 |
| 12.41 | 10.66 | 1.50 |
| 14.48 | 10.79 | 1.59 |
| **(b)** Corrected winds in the validation set | | |
| ASCAT-A and ASCAT-B wind speeds [m/s] | Mean values of the corrected winds [m/s] | SDD between the corrected and ASCAT winds [m/s] |
| 4.14 | 5.93 | 1.14 |
| 6.21 | 6.97 | 0.92 |
| 8.28 | 8.16 | 0.96 |
| 10.34 | 9.39 | 1.26 |
| 12.41 | 10.70 | 1.49 |
| 14.48 | 11.23 | 1.30 |
| **(c)** OSCAT-2 2DVAR winds (testing set) | | |
| ASCAT-A and ASCAT-B wind speeds [m/s] | Mean values of the 2DVAR winds [m/s] | SDD between the 2DVAR and ASCAT winds [m/s] |
| 4.14 | 3.06 | 1.86 |
| 6.21 | 5.22 | 1.95 |
| 8.28 | 7.43 | 1.84 |
| 10.34 | 9.01 | 2.19 |
| 12.41 | 10.46 | 2.37 |
| 14.48 | 11.04 | 2.04 |
| **(d)** OSCAT-2 observational winds (testing set) | | |
| ASCAT-A and ASCAT-B wind speeds [m/s] | Mean values of the observational winds [m/s] | SDD between observational and ASCAT winds [m/s] |
| 4.14 | 8.22 | 2.17 |
| 6.21 | 10.02 | 2.21 |
| 8.28 | 11.96 | 2.15 |
| 10.34 | 13.32 | 1.99 |
| 12.41 | 14.82 | 1.70 |
| 14.48 | 15.50 | 1.67 |

winds are not perfect and OSCAT is not perfectly collocated with ASCAT. Such uncertainty in ASCAT also has the tendency to flatten the red curves in Fig. 3.

In Fig. 4, the distributions of wind speed of the OSCAT-2 observational wind speed, OSCAT-2 2DVAR speed, collocated ASCAT speed and that of the SVM-corrected speed are displayed for the testing set.

From Fig. 4a, the blue curve indicates rain-affected OSCAT-2 winds are elevated and skewed to higher speeds, peaking at around 12 m/s. They also deviate from the corresponding 2DVAR speeds (purple) as well as the collocated ASCAT winds (green). Similar to the latter two, the

SVM-corrected winds (lighter blue) peak at a similar speed around 8 m/s. This is also consistent with Fig. 1a. Moreover, note that the 2DVAR wind distribution extends to the lowest speeds and deviates more than the corrected winds from ASCAT observations. Anyway, the corrected winds show a very similar shape to the ASCAT distribution, proving the effectiveness of the SVM. Figure 4b demonstrates the speed errors defined as the differences with respect to the ASCAT observations. Consistent with (a), the errors distribute more symmetrically and over the smallest range for the corrected winds. The more Gaussian-like features of this speed error as compared to the other groups can be more easily observed

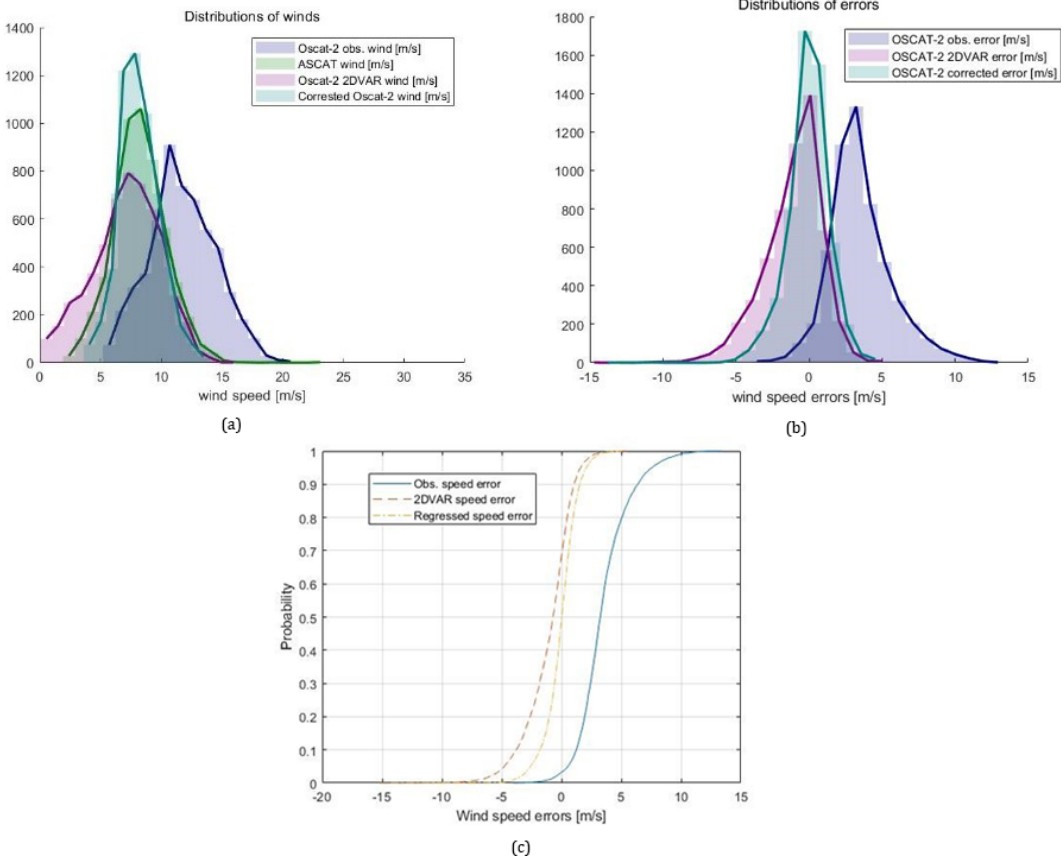

**Figure 4.** Distribution of the different wind speeds **(a)**, speed errors with the reference from ASCAT **(b)** and the cumulative distribution function (CDF) of speed errors **(c)** corresponding to the testing set.

from (c) where the cumulative distribution function (CDF) is obtained. In the figure, the blue, red and yellow lines are the CDFs of observational, 2DVAR and regressed speed error, respectively. Except for the most symmetric feature of the yellow curve in bias, about 90 % of the values lay between −2.0 and 2.0, which indicates again that the corrected winds are close to the ASCAT observations. In addition to Fig. 4, Fig. 5 demonstrates in detail and directly from the data that the statistics have been improved after SVM corrections.

Figure 5 is plotted from the testing set, where the horizontal and vertical axes are wind speed of ASCAT and that of observational and corrected OSCAT-2 speed in m/s for (a), (b) and (c), (d) respectively. Moreover, in (a) and (c), depicted in the colour bar, as functions of the horizontal and vertical speeds, are the average values of differences of speed from the vertical minus horizontal axis in corresponding bins. In (b) and (d), the colour represents WVC density in a bin. In (a), it can be observed that deviations from the C-band-accepted collocations due to rain vary with the reference wind speeds in a similar linear way, while for each wind speed there are multiple differences induced by rain. This is consistent with the quasi-linear relationship between Joss and rain rates in Fig. 2, and explains that such second-

order (speed difference vs. speed) relations involving multiple parameters (rain, wind and wind–rain correlations) cannot be corrected by simple linear methods. Meanwhile, in (b), the corresponding density of samples indicates non-uniform characteristics of the distribution of the differences for each reference speed (horizontal axis), implying skewed error distributions. At the same time, in (c) and (d), it can be seen that by SVM corrections, most of the differences are corrected, while (d) shows more evenly distributed difference patterns for the moderate wind speeds, where rain contamination effects appear better resolved, implying more uniform and normal difference values. This goes along with the distribution of corrected OSCAT winds slightly skewed away from the diagonal; this may be due to the lack of samples in higher wind speeds.

## 4.2 Spatial consistency of corrected winds

In this section, to obtain a spatial view of the results, figures of the collocated data on a randomly selected date (22 May 2017) are provided in Fig. 6, where (a) shows the wind speed of OSCAT-2 in both QC-I and QC-II collocations, and that of the rest of the FAE set. The same set is

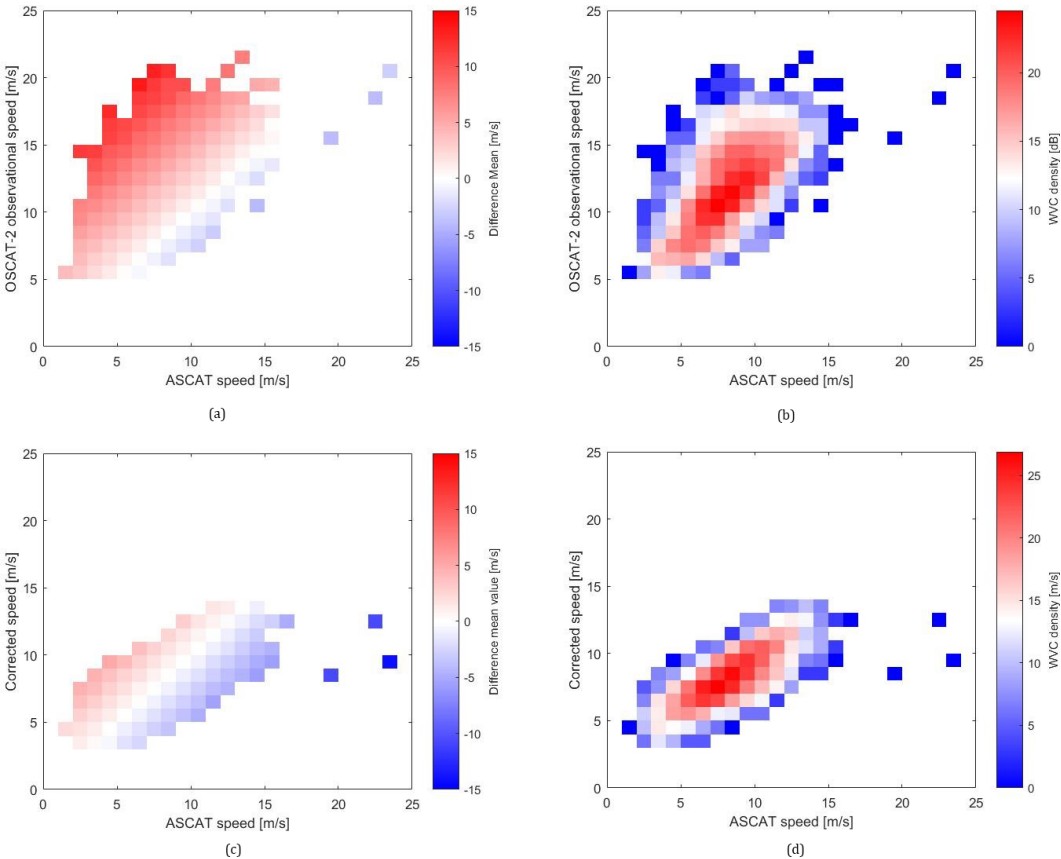

**Figure 5.** Mean values in 1 m/s bin of winds against mean difference values in the same bin of vertical minus horizontal values (colour) **(a)** and **(c)**, and sample WVC density **(b)** and **(d)**.

displayed in (b) but where the FAE OSCAT-2 wind speeds are from the SVM corrections. In (c), the regressed wind is replaced by the ASCAT-accepted winds. Furthermore, data in Fig. 4 are without GPM collocations, and the SVM winds are retrieved directly from the model established in Sect. 3.2.

In Fig. 6, the abscissas are longitudes, while the ordinate represents latitudes, and both are in degrees. Then the colour bars indicate wind speeds in m/s, where the ascending and descending tracks are displayed together, with latter observations obtained replacing the former ones. It can be observed that the colour red in (a) is suppressed in (b), while (b) is also more consistent with (c) than (a) is. This can be directly observed from (d), with the corrected wind locations from (e). Panel (f) shows a generally accepted correction in this region with speed higher than 12 m/s overestimated. Similar trends can also be noted in regions becoming much bluer, especially in cases that can be found near the red regions. *Nota bene*: the higher wind regions with speed larger than 15 m/s have fewer samples and are also limited by the FA rule limiting Joss to −1.33 m/s, above which, the wind–rain tangling at higher speed cannot be well resolved. Moreover, a region with no GPM collocation, and thus not involved in training procedures, is selected from the data set generat-

ing Fig. 7 and is shown in Fig. 8 as a case to validate the SVM regression method proposed. Wind speeds from the collocation set in QC-I, QC-II FA and QC-II FAE OSCAT-2 speeds are shown (a), along with that of QC-II FAE substituted by the SVM regressed speed for rain (cloud) correction (b) and that from the ASCAT collocations in the C band (c). There are 674 WVCs in Fig. 7, with 13 FAE values, and the observation time ranges from 09:19 to 09:24 UTC. Furthermore, the simultaneous image obtained around 09:20 is applied as reference from band 11 of Himawari-8 satellite at a medium infrared (MIR) wavelength of 8.6 μm from the Japan Aerospace Exploration Agency (JAXA).

In Fig. 7, the FAE set is distributed in the lower half in (a), where the colour is darker in red and lighter in white, implying the existence of a wind front. After the correction, a more consistent set of wind speeds north of the front is obtained. In addition, rain clouds can be seen from (d) between 7–9° N, with blue regions representing lower brightness temperatures (BTs) and high probability of rain, where rain correction effects can be observed as well considering (a)–(c). This further confirms the necessity of inclusion of 2DVAR Joss for wind correction in case of rain. Although slightly overcorrected wind speeds occur in (b) around about 8° N,

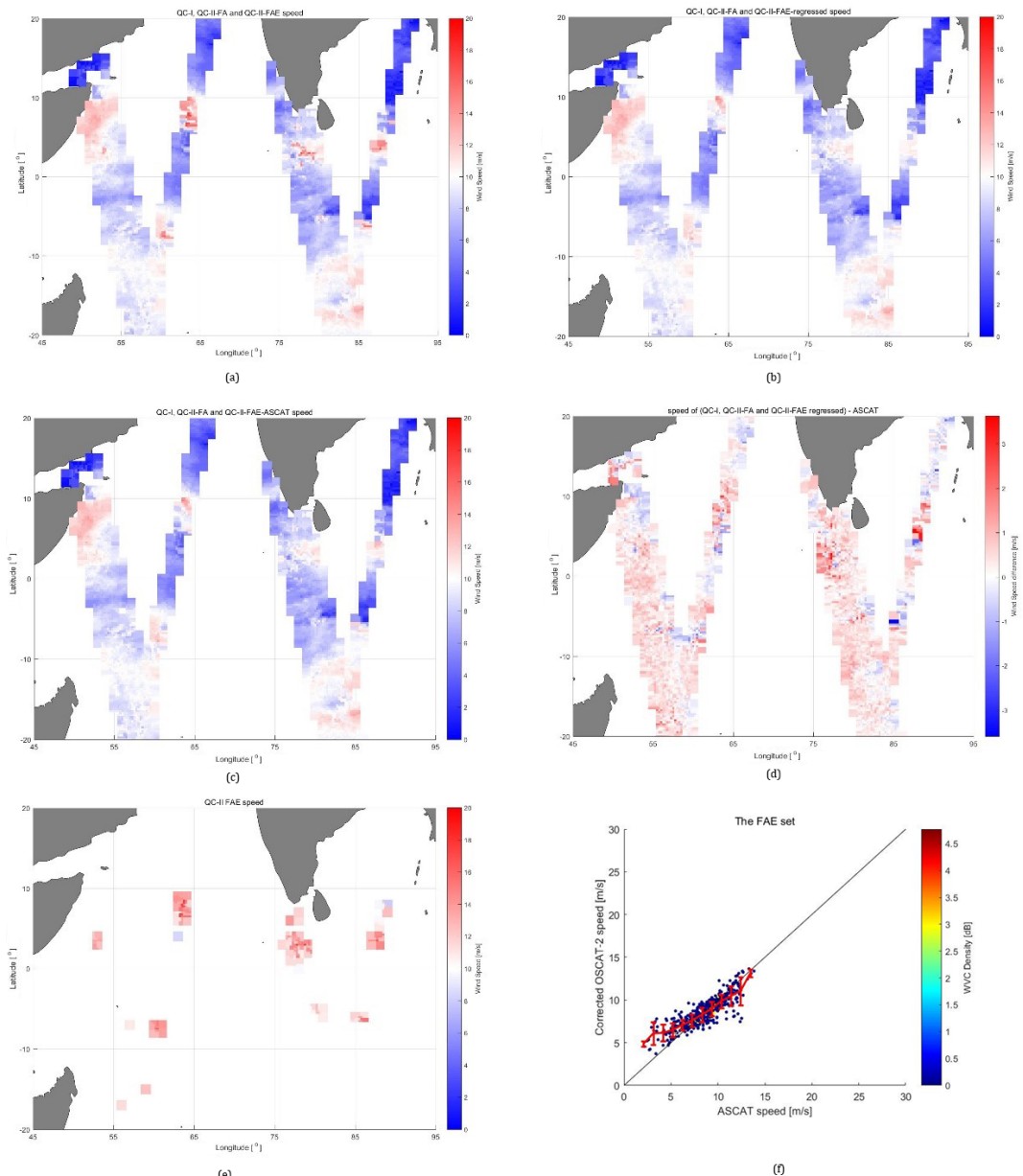

**Figure 6.** OSCAT-2 speed (m/s, in colour bars) for QC-I collocation set FA and FAE in the QC-II set **(a)**, and that of the QC-I, QC-II FA set when the FAE values in QC-II are replaced SVM regressed speeds **(b)**; then the FAE wind speeds are substituted by collocated ASCAT-A and ASCAT-B speeds **(c)**. Panel **(d)** shows the differences of speeds in **(c)** with their corresponding ASCAT speed, and **(e)** indicates the FAE location, while **(f)** shows the statistics of the corrected wind with ASCAT wind.

it can be observed that (b) and (c) are more similar than (a) and (c), demonstrating the consistency between the SVM-regressed OSCAT and accepted ASCAT wind speeds. This can be further observed from WVCs between 9–10° N, 175–176° E, where (d) shows somewhat elevated BT of clouds, illustrating the effectiveness of the method proposed for such regions. More detailed statistics are shown in Fig. 8.

It can be seen from Fig. 8 that higher wind due to rain is suppressed by the method proposed, while for higher

wind speed around 12 m/s, the SVM-regressed winds become somewhat less consistent with ASCAT *truth*, as discussed in the previous section. The effectiveness of the SVM-regressed winds is further confirmed by the data in Fig. 8, as they have not been applied in the derivation of the SVM.

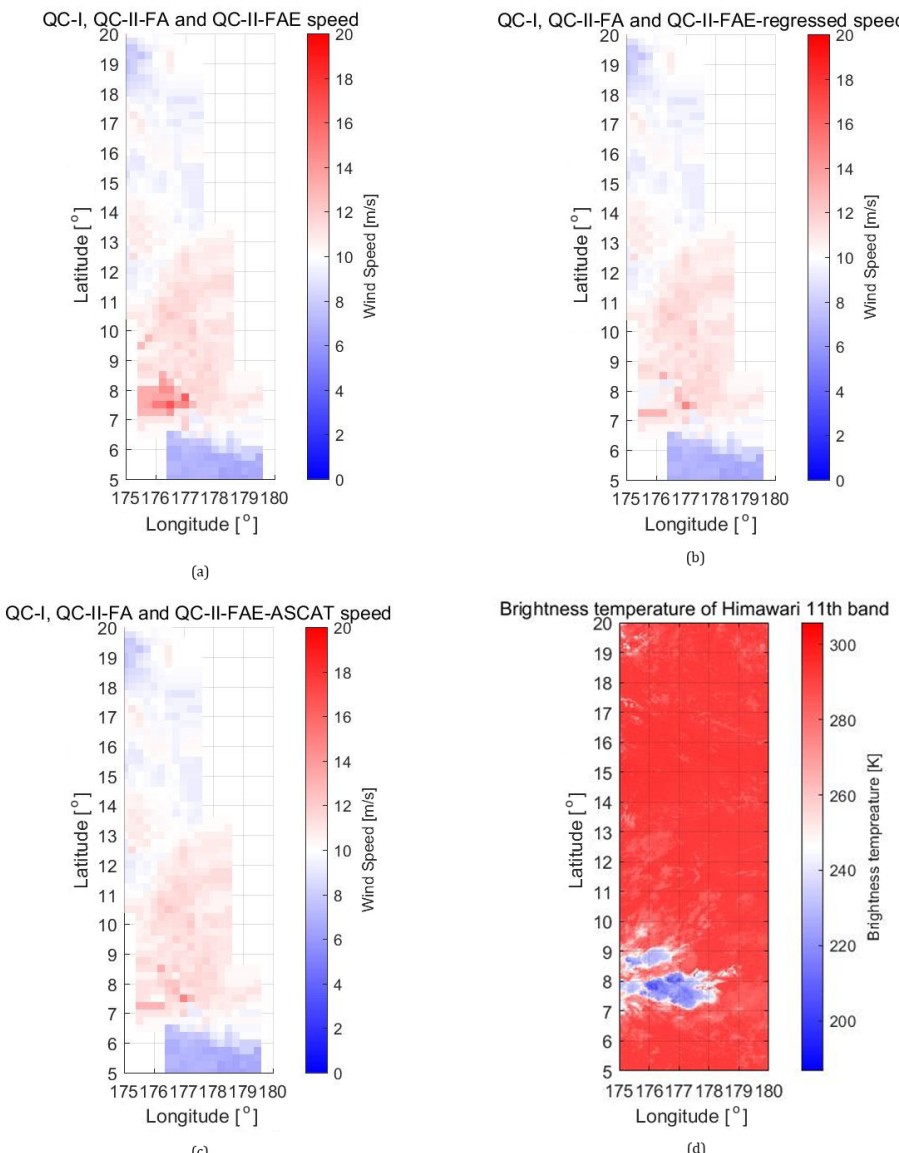

**Figure 7.** Wind speed of the QC-I, QC-II FA and QC-II FAE (**a**), with the FAE set replaced by the SVM regressed speed (**b**) and by speeds from their ASCAT collocations (**c**), with the synchronous MIR (**e**) images from Himawari-8, where the green rectangle indicates the region in panels (**a**), (**b**) and (**c**).

**Table 3.** Inputs and outputs for the SVM of rain classification and regression.

| Inputs | Output | | Values of output for training | |
|---|---|---|---|---|
| | Rain classification SVM | Rain regression SVM | Rain classification SVM | Rain regression SVM |
| OSCAT-2 MLE in dB space $\alpha$ from OSCAT-2 WVC OSCAT-2 2DVAR speed OSCAT-2 observational speed | Rain or no-rain class | Rain rates [mm/h] | GPM rain rate, 0 mm/h in no-rain WVC | GPM rain rates [mm/h] |

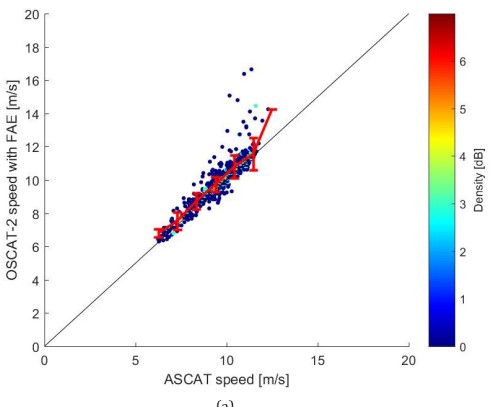
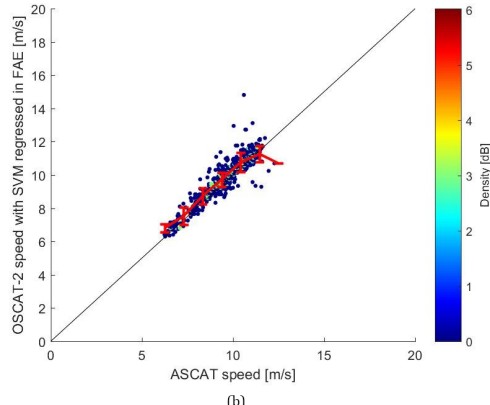

**Figure 8.** The FAE wind speed **(a)** and the corrected ones **(b)** against ASCAT wind speed in the data set of Fig. 6.

## 5   Discussion

Air–sea interaction in the vicinity of rain is complex and difficult to observe. In this research, the effect of rain in Ku-band wind scatterometry is explored for correction of retrieved wind under rainy conditions. The method employed is as follows: on the basis of the analysis of signatures induced by rain from parameters obtained during wind retrieval from scatterometers, rain effects are corrected as a function of these signatures. Specifically, for quantifying the heterogeneity induced by rain and its effect on the wind speed, the quality indicators MLE and Joss are analysed, with reference to the low-pass-filtered 2DVAR winds and collocated ASCAT winds (Xu and Stoffelen, 2020). Accepted C-band ASCAT winds (Vogelzang et al., 2011) are used as reference to identify the rain effects and form the basis of a correction after establishing a SVM. Results show that the correction is adequate, especially at speeds with abundant information in the Ku band to segregate wind and rain (under 12 m/s). The spatial consistency of the corrected winds with the ASCAT observational winds is identified as more similar compared to that with the 2DVAR winds. Subsequently, a case is provided with comparison to MIR images to check for rain occurrence. This confirms that the SVM method proposed is effective. Hereafter, rain information extraction from scatterometers is established. Following this, further analysis and discussion on the remaining uncertainties are given, with a view to improve in our future work.

### 5.1   SVM for rain identification and regression

For a view of uncertainties unresolved with wind–rain tangling in Ku-band wind scatterometry, SVMs in the same input for rain identification and regression are shown in Table 3.

The data set is the same as that for the wind correction, while the training target changed to GPM rain. The classification accuracies for both the training and testing sets of rain identification SVM are the same at 72 %. The results for

rain regression are shown in the following figure, where the correlation coefficient of the SVM-regressed and GPM rain rates for the training set and the testing set are both 0.47. Little skill for rain rate appears below 5 mm/h, while GPM produces more extreme rain rates >10 mm/h. The corresponding scatter plots of the regressed rain rates in the training set and testing set are depicted in Fig. 9.

From visualization of the classification results (details not shown), non-rainy WVCs are less often incorrectly classified than rainy WVCs. Higher 2DVAR speeds are well crowded and can be better discriminated in MLE, Joss and $\alpha$ to the correct class, while this is more difficult for lower 2DVAR speed WVCs. Light rain clouds have small effects on the wind observations. Correspondingly, Fig. 10a shows the distribution of rain rates from GPM (blue), SVM regression (purple) and that of the error defined as the GPM rain rate minus the regressed values (green). The corresponding CDF of error is shown in (b). In addition to Fig. 9, Fig. 10a shows in detail that SVM-regressed rain fails in capturing the non-convex feature in lower rain rate, and in prediction of higher rains. This may due to the L2 distance norm applied and lack of information as well as samples. For GPM rain above 10 mm/h, OSCAT-2 rain rates are rather randomly distributed and presumably lack skill. However, from (b) in Fig. 10, it can be observed that the error displays a feature of symmetry and steady increasing feature. And those within the range of [−2, 2] mm/h take 34 %, within [−5, 5] mm/h take about 80 %, consistent with the correlation coefficient value of 0.47. L1 distance (Manhattan distance), at the same time, including other sources of observation, with increasing number of samples may help improve the results. Xu et al. (2020a) find similar spread in rain products at the scatterometer spatial resolution, hence illustrating the applicability of the SVM rain product derived here.

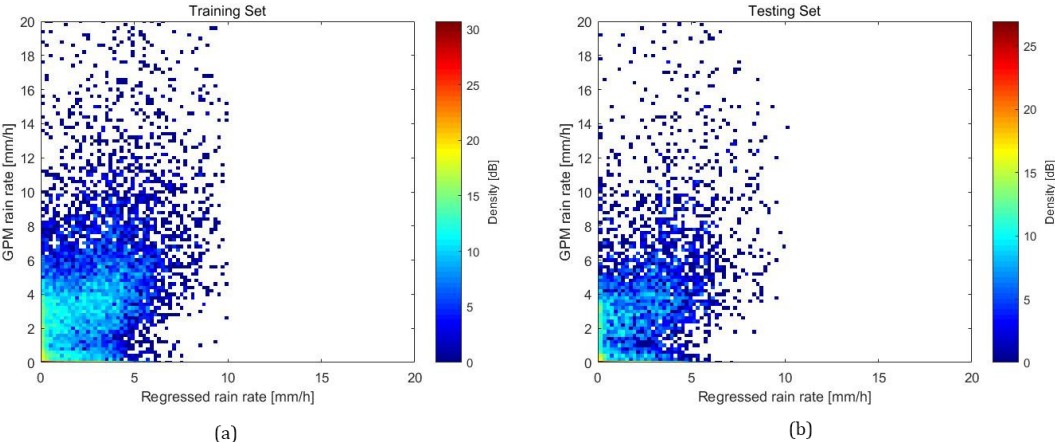

**Figure 9.** SVM regressed rains for training set **(a)** and validation set not involved in training **(b)**.

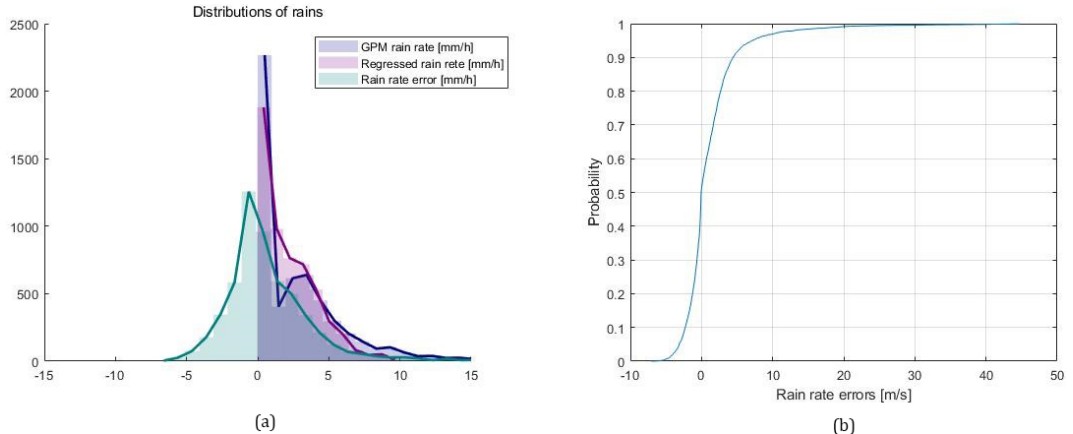

**Figure 10.** Distribution of GPM- and SVM-regressed rains with that of the error **(a)** and the corresponding CDF of the error **(b)**.

## 5.2 Conclusions and further research

Rain features in wind scatterometry in the Ku band can trigger QC rejections. These effects also provide opportunities to identify rain and perform wind corrections. The SVM method proposed performs well for medium and lower wind speeds, while the wind–rain tangling remains severe for higher wind speed. This can also be noted from the rain identification and regression SVMs in Sect. 5.1. For lower speeds, the change of values of parameters considered may be caused by different wind–rain interaction with the ocean surface that alters the sea state rather than only elevating the speed of wind due to rain cloud scattering that may be similar for C and Ku bands and hence missed here.

On the other hand, from the rain features in MLE and Joss, as well as the uncorrected speed, it can be seen that uncertainties can be introduced from the training parameters; the normalized MLE is designed to characterize errors that result in large deviations from the GMF for QC, but its accuracy depends on relative wind vector and azimuthal diversity of the NRCS views. The 2DVAR speed is derived by balancing errors in the observation space of a grid of WVCs and the NWP background, representing larger spatial scales; thus, they can be considered as lower-bound estimates of the *true* values, and uncertainties in the wind speeds can be different due to spatial heterogeneity. This may hamper the effectiveness of the rain screening ability of Joss. In order to bind those uncertainties for better results in SVM, extra observations for rain (clouds) can help, while higher spatial resolution is obtained in the next generation of scatterometers for simultaneous ocean surface wind and current measurements, for example, Chelton et al. (2019) and Du et al. (2021). OSCAT-2 and ASCAT collocations provided a unique opportunity to study rain effects in Ku-band scatterometers. Rain effects are rather transient in nature, where the moist convection timescale is about 30 min. This implies that updrafts, downdrafts and rain patterns in a WVC change very fast, and rather strict collocation criteria would be needed to resolve rain effects well. With WindRad on FY3E a combined C- and Ku-band scatterometer has been launched on 5 July 2021, which will pro-

vide parts of the swath with excellent azimuth diversity and both C- and Ku-band retrieval capability. Hence, this mission will be useful to further elaborate on this research.

Above all, the SVM can effectively represent the increasing effect of rain in elevating wind speeds as the true wind speed decreases showing the advantage of the ML method for such complex problems involving multiple interrelated variables. The method provides correction of deviations that are non-uniform and skew- to Gaussian-like features. This demonstrates the effectiveness of a ML method when used with representative parameters for addressing more complex problems. The corrected winds provide information previously lacking that is vital for nowcasting winds in the presence of moist convection and improving initialization of NWP models in dynamic conditions. The rain regression in SVM indicates the potential of additional rain information observations for further exploration, as well as the promise of improved hybrid wind and rain estimation methods based on ML using physically meaningful parameters for the problem at hand.

## Appendix A: The mean values and standard deviations of differences

We discuss the comparison of two collocated groups of data, one of which is set as reference group. Then figures and values are obtained by grouping the reference data (depicted as the horizontal axis) and the other data set to be compared (vertical axis) into $i$ bins of the same sample number $j$. For the mean values of the reference data, $\text{Ref}_i$ (in tables, they are put in the first column), there is corresponding $\text{Ave}_i$ (in tables, as the second column) and standard deviation values (third column) calculated for the data to compare (in figures, as the vertical axis). Specifically, the following equations describe the calculation of the mean value $\text{Ave}_i$ and standard deviation of difference (SDD) $\text{Std}_i$:

$$\text{Ave}_i = \frac{\sum_{j=1}^{N_i} \text{Obv\_Value}_j}{N_i} \tag{A1}$$

$$\text{Std}_i = \frac{1}{N_i} \sqrt{\sum_{j=1}^{N_i} \left( \text{Obv\_Value}_j - \text{Ref}_i \right)^2}, \tag{A2}$$

where the value of the group to compare is Obv_Value.

*Code availability.* There is no code available, but for the experiments, it can be reproduced upon request.

*Data availability.* The ASCAT-A, ASCAT-B and OSCAT-2 wind products applied are available from the Royal Netherlands Meteorology Institute (KNMI) data distribution site: https://osi-saf.eumetsat.int/products/wind-products (EUMETSAT, 2021) TS16. The GPM rain products are from the Precipitation Process Center, the National Aeronautics and Space Administration (NASA), available at https://gpm.nasa.gov/data/directory (NASA, 2021) TS17. The Himawari image data are available from the Japan Aerospace Exploration Agency (JAXA) at https://www.eorc.jaxa.jp/ptree/index.html (JAXA, 2021) TS18.

*Author contributions.* XX contributed to methodology, experiment, analysis and original draft writing of this research. AS contributed to the conceptualization, methodology, analysis, reviewing and implementation of this research.

*Competing interests.* Some authors are members of the editorial board of *Atmospheric Measurement Techniques*. The peer-review process was guided by an independent editor, and the authors have also no other competing interests to declare.

*Acknowledgements.* The authors would like to thank the Royal Netherlands Meteorology Institute (KNMI), the European Organization for the Exploitation of Meteorological Satellites (EUMETSAT), the European Centre for Medium-Range Weather Forecasts (ECMWF), the National Aeronautics and Space Administration (NASA) and the Japan Aerospace Exploration Agency (JAXA) for the provision of the data products applied.

*Review statement.* This paper was edited by Marcos Portabella and reviewed by two anonymous referees.

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

## Remarks from the language copy-editor

CE1    Please note the slight edits.

## Remarks from the typesetter

TS1    Please confirm.

TS2    Please confirm throughout.

TS3    Thank you for the reply. Are there any variables that are vectors (other than the ones that are already formatted bold italic)? If so, these variables have to be bold italic as well. Please check throughout.

TS4    Please confirm.

TS5    Please confirm "Joss".

TS6    Please confirm.

TS7    Please confirm.

TS8    Please give an explanation of why this needs to be changed. We have to ask the handling editor for approval. Thanks.

TS9    Please confirm.

TS10   Please confirm the vectors and equations.

TS11   Please state explicitly which terms need to be changed with the "cost" functions. Thank you.

TS12   Please confirm.

TS13   Please confirm the removal of the references.

TS14   Please note that the skinny space is only used if the numbers consist of five or more figures according to our standards.

TS15   Please confirm the table.

TS16   Please confirm citation.

TS17   Please confirm citation.

TS18   Please confirm citation.

TS19   Please note that I added the information. I would kindly ask you to check if this is really correct. If this is only a paper, "[code]" would be incorrect.

TS20   Please confirm reference list entry for the data set and provide date of last access.

TS21   Please provide the city.

TS22   Please confirm reference list entry.

TS23   Please confirm reference list entry for the data set and provide date of last access.

TS24   Please confirm the year.

TS25   Please check the link.

TS26   Please note that the year has been adjusted in the text as well.

TS27   Please note that Jones et al., 1979 has been removed.

TS28   Please confirm reference list entry for the data set and provide date of last access.

TS29   Please confirm.

TS30   Please provide date of last access.

TS31   Please note that the year has also been updated in the text.

TS32   Please confirm the added information.

TS33   Please provide date of last access.

TS34   Please confirm reference list entry.

TS35   Please confirm added information.

TS36   Please check all Xu et al., 2020 references throughout the text and confirm if they are labelled correctly. Thank you.

TS37   Please provide an update if possible.