# Peer review of "Support vector machine tropical wind speed retrieval in the presence of rain for Ku-band wind scatterometry"

_Atmospheric Measurement Techniques, 2021_

## Referee Comment (RC1)

**Review of the manuscript "Support vector machine tropical wind speed retrieval in the presence of rain for Ku-band wind scatterometry" by Xu and Stoffelen**

This paper uses the support vector machine (SVM) method to correct the Ku-band scatterometer wind speed under rainy conditions. To achieve promising results, the authors first employ the newly developed quality indicator, namely Joss, to filter the false alarmed winds (i.e., rain-free winds rejected the nominal quality control procedure) in the rejected data set, and then apply the SVM method to map several input vectors (2DVAR wind speed, OSCAT retrieved wind speed, normalized Joss, and MLE) to the 'true' wind speed (using the ASCAT winds as reference). The simultaneous wind and rain estimation using SVM is also presented, which results are promising and may be beneficial to a variety of applications.

The contents of this paper are clear, and the steps of the methodologies are well described. In my opinion, the manuscript presents a notable improvement in the estimation of scatterometer wind speed under rainy conditions. My comments are listed as follows:

1. The observed radar cross sections may still contain useful wind direction information under light rain conditions. Why the wind direction is not considered in the SVM model?

2. The authors mostly present the results of rain-contaminated OSCAT winds. To better understand the advantages of the proposed method, I'd suggest including the statistical scores of QC-accepted data (for reference) in the beginning of the results section.

3. Though the statistical scores of the corrected OSCAT wind speed are better than those of nominal wind inversion (at rainy area). Fig. 3(a) and (b) clearly show that there is a scaling issue with the corrected wind speed versus ASCAT wind speed. If this cannot be eliminated, the corrected wind speed may be useless. More discussions on this aspect would motivate the users to use the corrected winds in their applications.

4. Similar to 3, Fig.8 does not show remarkable impact of the SVM method on the wind correction. It looks to me the nominal wind inversion already does a very good job in retrieving winds for this particular case.

5. The figures presented in the manuscript are quite obscure, such as Fig. 6. The authors are encouraged to improve the quality of those figures.

6. Line 15, replace 'eliminating' by 'mitigating';
7. Line 34, remove 'sized square';
8. Line 42, remove 'described above' to keep the manuscript be concise

9. Line 43, 'Combined retrievals ….(Li et al., 2014)'. This sentence is a bit obscure, please rephrase.
10. Line 84, 'it is indicated' rephrased as 'it indicates'
11. Line 98, '… supports NWP before obtaining monitoring … models.' The second half of this sentence is obscure, please rephrase.
12. Line 115, I believe in PenWP the measured NRCS rather than the simulated ones are used in the denominator of this equation.
13. Line 116, the whole denominator (not Kp) represents the NRCS variance.
14. Line 144, do you think the threshold can be applied to other scatterometers, such HY-2 satellite scatterometers?
15. Line 151, 'Enhance wind variability …' rephrased as 'Increased wind variability …'
16. Lines 161-163, 'In the established model, …, by rain'. Conclusive sentence should appear in the section of results or discussions.
17. Lines 166-167, what do you mean by 'non-linear features are linearized'?
18. Line 182, the symbol at the left side of Eq. (5) is inconsistent with Eq. (4).
19. Line 184, the expression (6) is quite informal.
20. Line 196, 'more optimal NRCS' rephrased as 'better'
21. Line 198, remove 'And'
22. Line 223, what does 'latter distribution' mean?
23. Line 226, what do you mean "By design," ?
24. Above line 259, it seems the captions in the second columns of (c) and (d) are reversed.
25. Line 305, 'colour dimension' rephrased as 'colorbar'
26. Line 310, 'more simple' with respect to what?
27. Line 333, 'In Figure 6, … both are in degree.' This sentence is unnecessary.
28. Line 340, 'GMP' rephrased as 'GPM'
29. Line 350, Fig. 7(d), why don't you show the same region as Figs. 7(a)-(c)?
30. Line 358-395, 'This further confirms … in case of rain'. Indeed, the authors didn't test other input vectors for the SVM method, right? How do you get such conclusion?
31. Line 408, could you explain what is the 'non-convex feature'?

---

## Author Comment (AC1)

Reply to the review of the manuscript "Support vector machine tropical wind speed retrieval in the presence of rain for Ku-band wind scatterometry"

Dear Editor and Reviewers,

Thanks a lot for your work and comments concerning our manuscript! Based on the suggestions in your letters, we revised it and in the following content, your comments are replied item by item, with corresponding modifications to the manuscript listed. The replies are marked in blue, while revisions in the text are shown using red highlight for additions, and strikethrough font for deletions.

We highly appreciate your time and considerations.

Thanks again and best regards!

Xingou Xu and Ad Stoffelen

**Anonymous Referee #1**

"This paper uses the support vector machine (SVM) method to correct the Ku-band scatterometer wind speed under rainy conditions. To achieve promising results, the authors first employ the newly developed quality indicator, namely Joss, to filter the false alarmed winds (i.e., rain-free winds rejected the nominal quality control procedure) in the rejected data set, and then apply the SVM method to map several input vectors (2DVAR wind speed, OSCAT retrieved wind speed, normalized Joss, and MLE) to the "true" wind speed (using the ASCAT winds as reference). The simultaneous wind and rain estimation using SVM is also presented, which results are promising and may be beneficial to a variety of applications."

"The contents of this paper are clear, and the steps of the methodologies are well described. In my opinion, the manuscript presents a notable improvement in the estimation of scatterometer wind speed under rainy conditions. My comments are listed as follows:"

**Response:** Thank you for your careful review! We appreciate your positive evaluation of our work.

1. The observed radar cross sections may still contain useful wind direction information under light rain conditions. Why the wind direction is not considered in the SVM model? Response: Wind direction is considered since the current 2DVAR MSS scheme has the implicit skill of finding a spatially representative wind direction not affected by rain and we now more clearly explain this. We found that "the speed component of Jos (the vector differences of observational winds to 2-DVAR wind), called Joss, is sensitive to rain and has a direct link to rain rates" as demonstrated in the paper "Improved Rain Screening for Ku-Band Wind Scatterometry" (Xu and Stoffelen, 2020). The current QC is successful in eliminating WVCs that are rain contaminated, which results in a good wind direction skill of 2DVAR, though somewhat spatially smoothed. Thus based on the factors above, wind direction is currently not included in the SVM model. The reference (Vogelzang et al, 2009) is newly included with further explaination and the sentences above equation (2) have been modified to: "The wind speed correction procedure employed here, hence does not change the 2DVAR analysis field, nor the selected wind direction at the rain-affected WVCs obtained during the elaborated 2DVAR Multiple Solution Scheme (MSS) (Vogelzang et al, 2018). The rain effect is estimated by the wind speed difference of the 2DVAR analysis wind speed f and the selected observational wind speed fs, corresponding to the wind direction obtained by 2DVAR (Xu et al., 2020a):".

2. The authors mostly present the results of rain-contaminated OSCAT winds. To better understand the advantages of the proposed method, I'd suggest including the statistical scores of QC-accepted data (for reference) in the beginning of the results section.

**Response:** Thanks for this suggestion. However, for the C-band collocated reference, the QC-II set is indeed the accepted set with reliable winds in the tropical regions, while for the Ku-band winds the associated rejections are potentially affected by rain, thus QC-II FAE are set as inputs to the SVM for Ku-band rain correction. Furthermore, the Missed Detection Rate (MDR) in the operational QC, accepted by both C- and Ku-band winds in QC-I, only constitutes a small amount of the observations (2,128 of 1,185,469 WVCs, 0.18%) (Xu and Stoffelen, 2021), so we don't think these QC-I WVCs will show rain features well. Besides, for the references of features of the 2DVAR, OSCAT-2 and ASCAT speeds that are not demonstrated in Figure 1, please also refer to Xu and Stoffelen in 2020.

3. Though the statistical scores of the corrected OSCAT wind speed are better than those of nominal wind inversion (at rainy area). Fig. 3(a) and (b) clearly show that there is a scaling issue with the corrected wind speed versus ASCAT wind speed. If this cannot be eliminated, the corrected wind speed may be useless. More discussions on this aspect would motivate the users to use the corrected winds in their applications. **Response:** In section 4.2 and the discussion part, consistent with this suggestion, we've pointed out that the correction at lower and higher speed requires improvement indeed and further research to this end is our next step, as expressed in the first paragraph of part 5. Glancing at the figures, there is indeed some more scaling required for the corrected wind with reference to the ASCAT *truth*. However, in addition to the improved statistics in table 2, we demonstrate in section 4.1 and 4.2 the effectiveness and applicability of the corrected winds, from which it can be observed that the scaling is a secondary issue, where the majority of the corrected speeds lign up well with ASCAT

4. Similar to 3, Fig.8 does not show remarkable impact of the SVM method on the wind correction. It looks to me the nominal wind inversion already does a very good job in retrieving winds for this particular case.

**Response:** We note that the number of WVCs with changes is limited, but changed WVCs show more spatial consistency. This is in line with the response to 3, where the

improved statistics of the corrected winds are discussed. Please note (d) in Figures 3 and 4, where the uncorrected observational winds show rather large distortion.

5. The figures presented in the manuscript are quite obscure, such as Fig. 6. The authors are encouraged to improve the quality of those figures.

**Response:** Thanks again, we now generated the figures in a better resolution.

6. Line 15, replace "eliminating" by "mitigating";

**Response:** We replace the word "eliminating" to "reducing much", as mitigating is a little vague for the work in the manuscript.

7. Line 34, remove "sized square";

**Response:** this is removed and the sentence is modified to "a WVC is a square of size  $25 \text{ km} \times 25 \text{ km}$ ".

8. Line 42, remove "described above" to keep the manuscript be concise; **Response:** this is removed according to this comment.

9. Line 43, "Combined retrievals ....(Li et al., 2014)". This sentence is a bit obscure, please rephrase.

**Response:** this sentence is rephrased as: "Combined retrievals of wind and rain are generally applying synchronous passive measurements from radiometers for rain in the scatterometer case (Stiles et al., 2010). While, in addition to rain, winds are retrieved from GPM (Li et al., 2014)."

10. Line 84, "it is indicated" rephrased as "it indicates" **Response:** this is modified according to this comment.

11. Line 98, "... supports NWP before obtaining monitoring ... models." The second half of this sentence is obscure, please rephrase.

**Response:** this sentence is rephrased as: "Research on observation errors, i.e., the deviations from the truth, together with the monitoring information obtained from differences between scatterometer winds and models, supports NWP." And the following "Among which" is altered to "Among the errors".

12. Line 115, I believe in PenWP the measured NRCS rather than the simulated ones are used in the denominator of this equation.

**Response:** Thanks again for this detailed notification. Although the equation as indicated in the content of the manuscript: (Stoffelen, A. and Anderson, D., 1997) suggests the NRCS either from the observed or the simulated ones, PenWP indeed computes MLE from the measured ones. We changed the manuscript.

13. Line 116, the whole denominator (not Kp) represents the NRCS variance.

**Response:** This sentence has been modified from: "Kpi represents the variance of NRCS within this WVC" to "Kpi is a dimensionless constant determined by instrument noise".

14. Line 144, do you think the threshold can be applied to other scatterometers, such HY-2 satellite scatterometers?

**Response:** Yes, we think so, as operational Ku-band pencil-beam scatterometers are very similar in observing geometry, while for HY-2D, although in an inclined orbit, the NRCS rain effects are similar in the tropical regions.

15. Line 151, "Enhance wind variability ..." rephrased as "Increased wind variability ..."

**Response:** We think that the word "enhanced" shows a more direct causal link between the wind variability and the "enhance" of MLE, so may we keep this sentence as it is.

16. Lines 161-163, "In the established model, ..., by rain". Conclusive sentence should appear in the section of results or discussions.

**Response:** Thanks for this point! We now modified this sentence to be not conclusive: "In the established model, corrected winds closer to the observed C-band winds may be obtained for rain-affected Ku-band WVCs by eliminating non-Gaussian errors within a WVC caused by rain. Moreover, the SVM model, when established, could be applied for Ku-band rejections." from "In the established model, corrected winds closer to the observed C-band winds can be obtained for rain-affected Ku-band WVCs eliminating non-Gaussian errors within a WVC caused by rain. Moreover, the SVM model, when established, can be applied for Ku-band rejections.", where the "can be" are replaced by "may be" and "could be" respectively.

17. Lines 166-167, what do you mean by "non-linear features are linearized"?

**Response:** When expressing a set of vectors in the space determined by support vectors, the non-linear separated data become linearly related (for classification in SVM this is more direct, as the data sets become able to get separated by planes instead of some distorted surfaces).

18. Line 182, the symbol at the left side of Eq. (5) is inconsistent with Eq. (4).**Response:** No, they are slightly different in that (5) is a specific case of (4), as described in the content. Thus are put in different expressions.

19. Line 184, the expression (6) is quite informal.

**Response:** This is an objective function, in this form to indicate the optimizing rule. A pair of brackets are included now for clarity (to minimize distance and minimize weights; for specific content, please see the reference).

20. Line 196, "more optimal NRCS" rephrased as "better" **Response:** this is modified according to this comment.

21. Line 198, remove "And"

Response: the connection word "And" has been modified to "While".

22. Line 223, what does "latter distribution" mean?

**Response:** It was to indicate the ASCAT distribution. The sentence has been modified from "while latter distribution appears nominal and similar to that in Figure 1(a)" to "as the ASCAT wind distribution appears nominal and similar to that in Figure 1(a)".

23. Line 226, what do you mean "By design,"?

**Response:** We designed the plots (c) in Figures 1 and 2 to be similar for comparison. "By design" is now changed to "In comparison".

24. Above line 259, it seems the captions in the second columns of (c) and (d) are reversed.

**Response:** Yes, they were, this is modified according to this comment.

25. Line 305, "colour dimension" rephrased as "colorbar"**Response:** this is modified according to this comment.

26. Line 310, "more simple" with respect to what?

**Response:** with reference to a non-linear method, but this is not a proper statement here and the word "more" is deleted.

27. Line 333, "In Figure 6, ... both are in degree." This sentence is unnecessary.**Response:** We think that the coordinates need to be explained, hence we keep this sentence.

28. Line 340, "GMP" rephrased as "GPM"**Response:** This is modified according to this comment.

29. Line 350, Fig. 7(d), why don't you show the same region as Figs. 7(a)-(c)? **Response:** This is modified according to this comment.

30. Line 358-395, "This further confirms ... in case of rain". Indeed, the authors didn't test other input vectors for the SVM method, right? How do you get such conclusion? **Response:** Good point; this has now been further clarified. The reason of inclusion of 2DVAR *Joss* is motivated in the theoretical part, when we set the model by determining the inputs, and this sentence is a further practical confirmation to support the theoretical motivation. From a conceptual point of view, the *MLE* test identifies the WVC NRCS sets that do not follow the wind GMF. Two main reasons have been identified for such discrepancy, which are 1) enhanced wind variability and 2) rain. *Joss* well indicates the presence of rain and can be used to segregate wind variability and rain cases. The manuscript now better motivates this assumption. The current PenWP QC test is rather liberal in terms of rain screening and not much rain remains in the accepted Ku-band winds (see also our response to comment 2). Hence running tests over this data set mainly provides statistical biases and noise, since no substantial physical rain effect remains. Corresponding explanations are now added between lines 55 and 70.

**31. Line 408, could you explain what is the "non-convex feature"?**

**Response:** The non-convex feature corresponding to the existence of a valley part in the purple curve, which brings in the possibility of local optimization.

**Thanks a lot again for your careful and detailed review!**

**References:**

Xu, X. and Stoffelen, A.: Improved rain screening for ku-band wind scatterometry, IEEE Transactions on Geoscience and Remote sensing, 58, 2494-2503, 10.1109/TGRS.2019.2951726, 2020

Xu, X. Stoffelen, A.: A Further evaluation of the quality indicator Joss for Ku-band wind scatterometry in tropical regions, IEEE International Geoscience and Remote Sensing Symposium (IGARSS) (accepted), 2021.

Stoffelen, A. and Anderson, D.: Scatterometer data interpretation: Measurement space and inversion, Journal of atmospheric 510 and oceanic technology, 14, 1298-1313, 10.1175/1520-0426(1997)0142.0.CO;2, 1997

Vogelzang, J, Stoffelen, A. Improvements in Ku-band scatterometer wind ambiguity removal using ASCAT-based empirical background error correlations. Q J R Meteorol Soc. 2018; 144: 2245–2259. https://doi.org/10.1002/qj.3349

**Anonymous Referee #2**

The authors have devised a machine learning technique to correct Ku-band scatterometer wind speeds in the presence of rain contamination using a supervised learning technique in which C-band scatterometer winds that are less impacted by rain are used as training outputs. The technique appears offer significantly improved wind speed biases on validation data that was not used to tune the algorithm. It is a valuable contribution to the literature.

Thanks a lot for reviewing and your positive evaluation of our work!

Specific comments:

1. There still appears to be room for improvement in the statistics of the corrected speeds (e.g. Figure 3) as compared to cited results obtained by others.

**Response:** The results cited are from different approaches with different features and sampling, as stated in the introduction part. We did not implement these methods and make comparisons. Nor did the cited studies compare to ASCAT, but to NWP models that do neither resolve the wind variability nor the rain that cause the distortions in case of moist convection The merit of the results from this manuscript is in the corrected value taken from observations as targets. From this point of view, together with the theoretical part reasoning the choice of inputs, based on the physical information included in them, this could be the best results in such approach yet obtained.

2. The equations for computing the statistics in Table 2 needs to be spelled out.

**Response:** An appendix is newly included for explaining this.

3. The rain regression results have such large errors that it is hard to ascertain the merit of doing the regression.

**Response:** Evidence has been added to ascertain the relative success of the rain regression. The regression of rain is put in the discussion part for a direct view of windrain information obtained in scatterometer observations. The sensors are not specified for rain, while it indeed well captures rain that cast effects on winds. And this fact is vital for the further discussion part in 5.2 following rain regression results. We introduced a reference from Liu et al. (2020) that presents the high variability of tropical rain products and their effect on collocation and verification, which helps to verify the success and relative merit of the SVM rain product. At the end of the paragraph around line 40, we include the sentence: "though the high spatial and temporal variability of rain generally challenges small collocation errors and high correlation between instantaneous rain data sets (e.g., Liu et al., 2020)".

Following this comment, we also added at the end of 5.1 the sentence: "Xu et al. (2020b) find similar spread in rain products at the scatterometer spatial resolution, hence illustrating the applicability of the SVM rain product derived here.", and modified the third sentence in 5.2 by pointing out the function of SVM in 5.1 from "This can also be noted from the rain identification and regression SVMs" to "This can also be noted from the rain identification and regression SVMs in 5.1". The corresponding content in the abstract is also altered. Besides, the function of the SVM proposed is further clarified in the methodological part, with the sentence "We seek to address and correct biases in Ku-band scatterometer wind retrievals due to rain." included around line 115.

4. It appears that only the low quality QC-II data is corrected for rain. Is this appropriate? **Response:** The acceptance of Ku-band winds is well quality controlled. The MLE QC threshold is typically 2 standard deviations and rejects 5% of WVCs, suggesting a success rate of about 80% for identifying non-nominal and variable WVCs, including rain in about a third of these cases. Though there is a Missed Detection Rate (MDR) as

measured by the collocation criteria and data in the operational QC, it only forms a small amount of the observations (2,128 of 1,185,469 WVCs, 0.18%) (Xu and Stoffelen, 2021), confirming the likely success of the MLE QC in identifying variable and rainy anomalies. At the same time, existing research indicates the FAR in the rejections are poorly correlated with rain and thus are not used. More specific descriptions are in the content below equation (2) in the manuscript. Furthermore, for other effects inducing QC rejections in the tropical regions after SST correction for Ku-band, local wind variability in the tropical region is generally introduced or companied by rain/rain clouds. Related discussions are in the content around line 430 of the modified manuscript. Moreover, following concerns of the first reviewer, we modified the manuscript between lines 55 and 70 to better describe the logic in our approach.

5. Might it be advantageous to correct WVCs with light rain that are moderately contaminated or for that matter to train the technique to apply small (or zero) corrections to uncontaminated data?

**Response:** On one hand, light rains are easily missed for a microwave radar operating in the mode as a wind scatterometer and good wind estimations can be obtained. This also implies little rain information is captured under such conditions. On the other hand, local wind variability can accompany such rain cases, also, local sea surface roughness can be altered by them. These are effects from/of rain, and if large enough, can trigger QC into the rejection category, which is the set employed in our research. Besides, as in the response on the 4th comment, the Missed Detection Rate (MDR) in the QC only forms a very small amount of the observations.

6. Table 1 lists the inputs. I suspect one could achieve better performance by also including the NRCS measurements themselves or mean values of those measurements for each azimuthal look as inputs to the SVM. Estimates of brightness temperature, if available, could also be helpful.

**Response:** The setting of the inputs for constructing the SVM model is motivated in the theoretical part, where it is argued that the selected parameters are paramount and probably sufficient to rain information observed in Ku-band wind scatterometry. As for

the direct input of NRCS measurements, on one hand, variance from rain clouds is extracted from the wind inversion procedure with reference to the wind GMF, which is well established, and for all NRCS within a WVC, we can also estimate a fractional (rain area) parameter, i.e.  $\boldsymbol{\alpha}$ . NRCS in a WVC in itself does not provide information on the heterogeneity nature of rain (clouds) or wind variability within it, hence in the presence of rain effect obscures the wind information.On the other hand, the wind information in the NRCS set projects onto the wind GMF, while rain and wind variability within a WVC project onto the MLE. Due to the 2DVAR property of providing a good wind direction estimate, the main rain effect is visible in the wind speed residual through Joss. Adding indeterminate parameters could in turn introduce uncertainty into the SVM model. To make this point more clear, we've modified the last sentence in the manuscript from "The rain regression in SVM indicates the potential of additional rain information observations for further exploration, as well as the promise of improved hybrid wind and rain optimization methods based on ML." to "The rain regression in SVM indicates the potential of additional rain information observations for further exploration, as well as the promise of improved hybrid wind and rain estimation methods, based on ML using physically meaningful parameters for the problem at hand.".

Brightness temperatures could bring further information on rain, if provided at sufficient accuracy and spatial resolution. eFurther information about the BTs and corresponding analysis maybe required before clear conclusions can be drawn.

**Thank you again for your elaborated work on this manuscript!**

**Reference:**

Xu, X. Stoffelen, A.: A Further evaluation of the quality indicator Joss for Ku-band wind scatterometry in tropical regions, IEEE International Geoscience and Remote Sensing Symposium (IGARSS) (accepted)